# A radar view of ice microphysics and turbulence in Arctic cloud systems

Jialin Yan <sup>1</sup>, Mariko Oue <sup>1</sup>, Pavlos Kollias <sup>1,2</sup>, Edward Luke <sup>1,2</sup>, and Fan Yang <sup>2</sup>

**Correspondence:** Jialin Yan (jialin.yan@stonybrook.edu)

Abstract. Ice microphysical processes are inherently complex because of their sensitivity to temperature and humidity, the diversity of ice crystal habits, and their interaction with supercooled liquid water (SCL) and turbulence. Long-term surface-based radar observations have been systematically used to unravel the different processes that affect ice particle growth. In this study, we present a statistical analysis of 6.5 years of Ka-band radar observations in Arctic cloud systems, combined with thermodynamic profiles derived from radiosonde measurements. For the first time, ice particle growth and sublimation—diagnosed from vertical gradients of radar reflectivity and mean Doppler velocity—are systematically mapped across a broad range of temperature and moisture conditions. These vertical gradients correspond closely to saturation levels relative to ice and exhibit a strong temperature dependence in supersaturated regions. Notably, distinct signatures near -15°C are indicative of dendritic growth. Turbulence, quantified via the eddy dissipation rate (EDR), is most frequently observed in regions containing SCL. The co-occurrence of SCL and elevated turbulence results in significantly enhanced ice particle growth compared to conditions in which either is present alone.

This work provides new observational constraints that are critical for improving the representation of ice microphysics in atmospheric models.

## 1 Introduction

15 Ice clouds play a crucial role in Earth's radiation budget (e.g. Liou, 1986; Stephens et al., 1990; Lynch et al., 2002), and hydrological cycle since 70% of global precipitation originate from the ice phase (e.g. Heymsfield et al., 2020; Khain et al., 2015; Mülmenstädt et al., 2015). Ice particles can take many forms depending on the environment and the impact of complex microphysical processes. Accurately representing these processes in numerical models remains challenging due to the limited understanding of how ice particles evolve under different thermodynamic and dynamic conditions (e.g. Khain et al., 2015; Grabowski et al., 2019).

Ice particles develop different habits under varying environmental conditions primarily through the deposition of water vapors (e.g. Korolev et al., 1999; Bailey and Hallett, 2009; Baker and Lawson, 2006; Kikuchi et al., 2013). Among these, dendritic crystals have attracted particular attention due to their maximum depositional growth rates (e.g. Takahashi, 2014; Korolev, 2007) and fragile structure, which make them prone to break-up (e.g. Schwarzenböck et al., 2009; Rangno and

<sup>&</sup>lt;sup>1</sup>School of Marine and Atmospheric Sciences, Stony Brook University, Stony Brook, NY, USA

<sup>&</sup>lt;sup>2</sup>Environmental Science and Technologies Department, Brookhaven National Laboratory, Upton, NY, USA

Hobbs, 1998; Takahashi and Fukuta, 1995; Choularton et al., 1986; Vardiman, 1978). Secondary ice production (SIP), the formation of atmospheric ice requiring pre-existing ice particles, is also a key microphysical process in clouds (e.g. Kanji et al., 2017; Korolev and Leisner, 2020). More than six mechanisms have been identified through laboratory and field studies, but the Hallett-Mossop riming-splintering process remains the only one widely represented in models, typically occurring between -3 and -8 °C (Korolev and Leisner, 2020). Recent remote sensing observational evidence also suggests that secondary ice production frequently occurs at temperatures warmer than -10°C (e.g. Kumjian et al., 2020; Luke et al., 2021; Oue et al., 2018; Korolev et al., 2020). Although significant progress has been made through laboratory experiments, aircraft observations, and radar case studies, to our knowledge, a systematic understanding of how environmental conditions shape ice microphysics in natural clouds, particularly from a long-term observational perspective, is limited.

In addition to atmospheric thermodynamic conditions, turbulence has also been recognized as a factor influencing ice nucleation and particle growth, yet observational studies investigating this interaction in natural cloud systems are extremely limited. Terrain-induced flow modifications result in turbulence and vertical updrafts, which significantly shape the ice microphysical processes (e.g. Garrett and Yuter, 2014; Houze and Medina, 2005; Ramelli et al., 2021). Recent statistical analysis from 15-months of radar observations has revealed that the increased turbulence, quantified by the eddy dissipation rate (EDR), enhances aggregation and riming in Arctic low-level mixed-phase clouds (Chellini and Kneifel, 2024). However, this analysis does not account for the influence of humidity and temperature at each observation point, which may also play an important role.

The U.S. Department of Energy (DOE) Atmospheric Radiation Measurement (ARM) North Slope of Alaska (NSA) observatory (Verlinde et al., 2016) provides multi-year ground-based observations for ice clouds including Ka-band ARM Zenith-pointing Radar (KAZR) measurements, radiosondes, and ceilometer. KAZR provides much higher sensitivity to small cloud droplets and ice crystals, allowing clear detection of MDV deceleration when large precipitation particles coexist with smaller hydrometeors, compared to longer-wavelength radars (X, C, S bands) (Kollias et al., 2020). The objective of our study is to characterize ice microphysical processes (e.g. depositional growth and sublimation) and in-cloud turbulence, based on long-term KAZR observations under a wide range of environmental conditions derived from radiosondes and examine their relationships. The presented analysis aims to improve our understanding of ice microphysics in Arctic clouds and provide observational constraints for advancing the representation of ice microphysics in models.

#### 2 Methodology

#### 2.1 Data

The observations in this study are collected from the DOE ARM NSA (71°19′22.8″ N, 156°36′54″ W) atmospheric observatory from January 2013 to May 2019, which is the same dataset as used by Luke et al. (2021). This dataset provides a continuous and sufficiently long record for deriving robust statistics. Radar reflectivity and mean Doppler velocity (MDV) from the Ka-band ARM Zenith Radar (KAZR, Feng et al., 2011) are used, while temperature and moisture data from the radiosonde measurements (Keeler et al., 2002) are utilized to characterize the environmental conditions. The vertical and time resolu-

tions of the KAZR data are 30 meters and 3.7 seconds, respectively. Detailed descriptions of the KAZR radar and radiosonde instruments used at the NSA site are extensively covered in previous studies (Kollias et al., 2020; Luke et al., 2021).

To ensure adequate representativeness of the environmental conditions surrounding the radar profile at a given time, we only analyze KAZR radar observations occurring within +-15 minutes time and 4 km horizontal distance of a corresponding radiosonde measurement as the balloon ascends as shown in an example case in Fig. 1, following the approach used in (Luke et al., 2021). To ensure the focus remains on cold phase microphysical processes, we exclude the profiles containing warm clouds, with temperatures above 0°C according to the radiosonde profile. This selection remains a large dataset, with over 39 million valid radar observation points.

#### 2.2 Moisture conditions

The saturation ratio with respect to ice (Si) is calculated from temperature, relative humidity (RH), and pressure obtained from the radiosonde measurements. This is a good indicator to identify layers where ice hydrometers grow by water vapor deposition (Si > 1.0) or sublimate (Si < 1.0). In the example case on February 18, 2015, shown in Fig. 1, the layers with Si > 1.0 show good agreement with the cloud signals shown by radar reflectivity, except for altitudes below 3 km. The high reflectivity with Si < 1.0 below 3 km suggests that sublimation is dominant there.

Previous research indicates that radiosonde-observed high relative humidity is consistent with the presence of supercooled liquid droplets (SCL) at subfreezing temperatures. For example, Silber et al. (2021) used a threshold of RH above 95% to identify the SCL layer across a wide temperature range, while Luke et al. (2021) used a higher threshold of about 98% around -5 °C. In this study, we use the relative humidity threshold of 95% to indicate the potential existence of SCL, as it provides a broadly applicable criterion across the temperature range. We define three moisture regimes: SCL conditions (RH > 95%, Si > 1.0), which favors the existence of SCL, considering sufficient cloud condensation nuclei are available; ISO (ice supersaturated only) conditions (RH < 95%, Si > 1.0), where deposition is possible but the occurrence of liquid water is highly unlikely; and ice-subsaturated conditions (Si < 1.0), where neither liquid water nor depositional growth is expected.

As shown in Fig. 1, the cloud base heights observed by the ceilometer, which is the lowest altitude of liquid, exhibit good agreement with the base of the first SCL layer above the surface, identified by the radiosonde observations (i.e., RH exceeding 95%). This agreement supports our identification of SCL, with the ceilometer serving as an independent validation on the vertical placement of SCL inferred from radiosonde measurements. The second SCL layer recognized by RH > 95% is located near the top of the radar-detected cloud layer. SCL-topped clouds with ice precipitation below is a typical structure of mixed-phase clouds in Arctic (Morrison et al., 2012).

## 2.3 EDR calculation

80

85

The eddy dissipation rate (EDR), used to characterize turbulence intensity, is calculated from mean Doppler velocity observations from KAZR and horizontal wind data from radiosondes following the velocity time-series variance method described in Borque et al. (2016), which provides robust estimates in cases where the spread of the particle size distribution significantly affects spectrum width.

Figure 1. A case on February 18, 2015 (UTC) at the NSA site. (a) Radar reflectivity (dBZ) and (b) mean Doppler velocity (MDV) from KAZR within  $\pm 15$  minutes and 4 km of the radiosonde launch, plotted as time–height sections in UTC. (c) Radiosonde profiles of relative humidity and water vapor saturation ratio with respect to ice (Si). (d) In-cloud energy dissipation rate (EDR) from radiosonde and radar. In (a–b), dashed purple lines mark the top and bottom boundaries of radiosonde-identified supercooled liquid water (SCL) layers; for clarity, two closely spaced SCL layers separated by a thin non-SCL interval are shown as one continuous layer. Black points denote ceilometer cloud base height (Zhang et al., 1997). In (c), the vertical dashed red line marks RH = 95% (SCL threshold), and the vertical dashed blue line marks Si = 1. Colored shadings in (c) indicate moisture regimes: blue for the SCL condition (RH > 95%, Si > 1.0), purple for the ISO condition (RH < 95%, Si > 1.0), and orange for the ice-subsaturated condition (Si 

Figure 2. Median values of key variables as a function of relative humidity (%) and temperature ( $^{\circ}$ C), including (a) radar reflectivity, (b) mean Doppler velocity, (c) the vertical gradient of dBZ, and (d) the gradient of MDV. The black contour lines indicate different saturation levels with respect to ice (Si = 1.25, 1.0, and 0.75).

This analysis attempts to gain insights into the processes that govern ice particle growth and sublimation under different temperature and moisture conditions. Results show that the region near the ice saturation threshold (Si = 1.0) is associated with relatively larger reflectivity values (0-10 dBZ, Fig. 2a) and larger MDV values (generally > 0.7 m s<sup>-1</sup>, Fig. 2b) compared to both of ice supersaturated region (Si > 1.0) and sublimation region (Si < 1.0) at temperatures above -30°C. Notably, the regions where the dBZ gradients (Fig. 2c) or MDV gradients (Fig. 2d) approach zero closely coincide with Si  $\approx$  1.0. This alignment supports the expected physical interpretation: when Si < 1.0, sublimation dominates (negative dBZ gradient and MDV gradient), and when Si > 1.0, ice particle growth is favored (positive dBZ gradient and MDV gradient). These findings suggest that the vertical gradients of dBZ and MDV can serve as process indicators for ice growth and sublimation. This framework also explains why ice particles tend to reach their maximum size and/or density near Si = 1.0, under specific temperature conditions. Ice particles in the balanced region between growth and sublimation may have experienced growth

in super-saturated regions; and, in environments favorable for ice growth (Si > 1.0), particles are often still in the process of growing when observed, thus not arriving at their maximum size and density.

At lower temperatures (below -30°C), even when particles are in equilibrium (Si  $\approx 1.0$ ), both reflectivity (typically below -5 dBZ, Fig. 2a) and MDV (generally lower than 0.5 m s<sup>-1</sup>, Fig. 2b) remain relatively lower than those at warmer temperatures. Reduced heterogeneous nucleation in this cold temperature range may lower the number concentration and potentially allow existing individual particles to grow larger. However, we observe reductions in particle size and fall speed indicated by reflectivity and MDV, which suggests that the reduction in available SCL plays a more significant role in preventing particles from reaching larger sizes and higher densities, due to the absence of growth processes involving supercooled water (such as riming). However, it should be also noted that the Si estimation in this cold temperature has a large uncertainty.

In ice supersaturated regions (Si > 1.0), the vertical gradient shows a dependence on temperature (Figs. 2c-d). The dBZ gradient is relatively larger at around -5°C, -15°C, and -25°C compared to other temperatures, while MDV gradient is relatively lower at temperatures above -5°C and around -15°C compared other temperatures. Interestingly, these observed features align with the maximum mass growth rate of ice particles and their apparent density minima near -5°C and -15°C reported in vertical supercooled cloud tunnel studies (Takahashi et al., 1991; Takahashi and Fukuta, 1988). Except for the size or density growth rate, the particle shape transition may also be a significant source of change of scattering scattering properties as revealed by simulations (Tyynelä et al., 2023, and references therein). An increase in the reflectivity (dBZ) gradient and a decrease in the MDV gradient are observed around -15°C (Fig. 2c-d), which corresponds to significant minima in MDV (Fig. 2b). These phenomena have been frequently reported in previous observations (e.g. Schrom and Kumjian, 2016; von Terzi et al., 2022).

When examining the frequency distributions for each grid point in Fig. 2, multimodal patterns are observed in reflectivity and MDV. Figure 3 shows the normalized frequency distributions of reflectivity and MDV at different temperatures in the SCL (RH > 95%, Si > 1.0) and ISO (RH < 95%, Si > 1.0) conditions. Results show that the medians of reflectivity and MDV in SCL conditions are smaller than the medians in ISO conditions. It is consistent with suggestions from Fig. 2 that ice particles in the ISO conditions may have experienced the growth in SCL conditions for balanced regions (Si  $\approx$  1.0), and thus their reflectivity and MDV are larger than those in SCL region. In addition, the smaller MDV in the SCL condition can also be partly due to the presence of SCL droplets coexisting with primary ice particles, in which situation MDV could be smaller than the situation where ice particles exist only.

The reflectivity in the SCL condition is multi-modal as shown in Fig. 3a for temperatures warmer than -15°C, possibly because the reflectivity is also affected by differences in their growth processes determined by different particle trajectories. The population with weaker reflectivity suggests the dominance of small pristine ice crystals. The other population with larger reflectivity suggests the dominance of larger particles such as large pristine ice crystals, aggregates, and rimed particles. This is consistent with previous studies that showed active aggregation in the dendritic growth zone (Bechini et al., 2013; Andrić et al., 2013), and active riming at around -10°C (Takahashi et al., 1991). In addition, we also suggest that at around -15°C, the size of ice particles may have dependence on its duration time in its favorable conditions because of its slow fall speed indicated by low MDV.

Notable multimodality in MDV under the SCL condition is shown at temperatures warmer than -5°C in Fig. 3b, where there are two populations with MDV at around 1.2 m s<sup>-1</sup> and 0.0 m s<sup>-1</sup>. The slower MDV population indicates supercooled liquid droplets, corresponding to the reflectivity peak at about -25 dBZ in Fig. 3a. The MDV reaches its local maximum value at about -6 °C, while it attains its local minimum value near -15 °C for both SCL and ISO conditions. The decrease in MDV for the SCL condition is more pronounced than that for the ISO condition.

Figure 3. Normalized frequency distribution of reflectivity (dBZ, panels (a-b)) and mean Doppler velocity (MDV,  $m s^{-1}$ , panels (c-d)) as a function of temperature (°C) under different moisture conditions. Panels (a) and (b) correspond to SCL conditions, while panels (c) and (d) correspond to ISO conditions. The dashed black and red lines represent the median profiles for SCL conditions and ISO condition, respectively. The blue dashed line indicates the reference line for MDV = 0  $m s^{-1}$ . The color shading indicates the normalized frequency of observations within each temperature-MDV or temperature-reflectivity bin.

## 3.2 Microphysics in Dendritic growth regions

175 To further explore the particle size dependence in the observed patterns in the dendritic growth regions, Fig. 4 examines the dBZ gradient and MDV gradient as a function of dBZ and temperature for SCL conditions and ISO conditions, In SCL conditions, Fig. 4a clearly shows the significant increase in dBZ gradient at around -15°C at reflectivity from -40 to 10 dBZ. The relative small and even negative MDV gradient is significant at smaller reflectivity < -28 dBZ. For ISO conditions, the small MDV gradients coexist with large positive dBZ gradients (>30 dBZ km<sup>-1</sup>, Fig. 4c) when reflectivity < -5 dBZ. The small MDV gradient (< 0.2 m s<sup>-1</sup> km<sup>-1</sup>) at around -15°C extends to reflectivity of 20 dBZ, while the large dBZ gradients (> 20 dBZ 180 km<sup>-1</sup>) are limited to small reflectivity 

**Figure 4.** Median values of dBZ gradient and MDV gradient as a function of temperature (°C) and reflectivity (dBZ) under different conditions: (a) SCL conditions: dBZ gradient (dBZ km<sup>-1</sup>). (b) SCL conditions: MDV Gradient (m s<sup>-1</sup> km<sup>-1</sup>). (c) ISO conditions: dBZ Gradient (dBZ km<sup>-1</sup>). (d) ISO conditions: MDV Gradient (m s<sup>-1</sup> km<sup>-1</sup>).

-20°C. This feature may be explained by the presence of relatively large and fast-falling snow particles indicated by large reflectivity, which enhances riming and aggregation. In addition, its coincidence with the temperature range of -10 to -20 °C further supports the aggregation, as dendrite in this regime are known to aggregate more efficiently (Lamb and Verlinde, 2011). Under ISO conditions, strongly positive MDV gradients are observed at high reflectivity within the temperature range of -16 to -25°C. According to Fukuta and Takahashi (1999), the fall speed of ice particles arrives a local minimum at around -22°C and a local maximum at around -20°C when the growth time is sufficiently long, which may contribute to the observed behavior. A less pronounced pattern of MDV gradient minima and reflectivity maxima is observed around -22°C for SCL conditions for small particles (around -40 dBZ) in Fig. 4 c-d. This also suggests that ice particle type transition is happening around that temperature.

## 3.3 The relationship between turbulence, SCL, and ice microphysics

220

225

Figure 5a illustrates the distribution of the turbulence, represented by the EDR, under varying temperature and moisture conditions. A threshold of  $10^{-5} \,\mathrm{m}^2 \,\mathrm{s}^{-3}$ , corresponding to the local minimum between two peaks in the EDR distribution (Fig. 5b), was used to separate turbulence regimes in the local context. Hereafter, we refer turbulent condition as EDR >  $10^{-5} \,\mathrm{m}^2 \,\mathrm{s}^{-3}$  (high EDR), and non-turbulent condition as EDR  $\leq 10^{-5} \,\mathrm{m}^2 \,\mathrm{s}^{-3}$  (low EDR). As shown in Fig. 5a, most of the medians of

Figure 5. (a) Median values of EDR as a function of relative humidity (%) and temperature ( $^{\circ}$ C), the black contour lines indicate different saturation levels with respect to ice (Si = 1.25, 1.0, and 0.75); (b) PDF of EDR under SCL conditions (blue line), ISO conditions (orange line), and Si < 1.0 conditions (green line). Vertical distribution of in-cloud EDR under (c) SCL conditions and (d) ISO conditions.

the EDR in SCL conditions are of values greater than  $10^{-5} \, \mathrm{m^2 \, s^{-3}}$ , and larger than those in ISO and Si < 1.0 conditions. The PDFs of EDR also show that EDR values in SCL conditions are relatively larger, suggesting stronger turbulence therein (Fig. 5b). The strong correlation between high EDR and the SCL condition may be related to multiple processes. On the one hand, liquid droplet formation and growth in the mixed-phase clouds can enhance turbulence via the latent heat release. In addition,

**Table 1.** Median and skewness of dBZ and MDV gradients under six thermodynamic-turbulence conditions. Units: dBZ gradient in dBZ  $km^{-1}$ , MDV gradient in  $m s^{-1} km^{-1}$ , EDR in  $m^2 s^{-3}$ .

| Condition         | Variable     | Median | Skewness |
|-------------------|--------------|--------|----------|
| SCL,              | dBZ Gradient | 12.99  | 1.00     |
| High EDR          | MDV Gradient | 0.96   | -0.41    |
| SCL,              | dBZ Gradient | 8.17   | 0.96     |
| Low EDR           | MDV Gradient | 0.25   | -0.82    |
| ISO,              | dBZ Gradient | 9.90   | 1.11     |
| High EDR          | MDV Gradient | 0.48   | -0.59    |
| ISO,              | dBZ Gradient | 6.34   | 0.33     |
| Low EDR           | MDV Gradient | 0.14   | -0.55    |
| Ice-subsaturated, | dBZ Gradient | -9.44  | 0.61     |
| High EDR          | MDV Gradient | 0.18   | -0.74    |
| Ice-subsaturated, | dBZ Gradient | -6.45  | -0.71    |
| Low EDR           | MDV Gradient | -0.04  | -0.61    |

stronger radiative cooling above the liquid cloud top can also enhance turbulence therein (e.g. Lonardi et al., 2024). On the other hand, turbulence may play an active role in maintaining humidity conditions favorable for the SCL formation (Morrison et al., 2012; Korolev and Field, 2008), potentially creating a positive feedback.

We further analyze the normalized frequency of the EDR as a function of normalized distance from cloud top under SCL and ISO conditions, as shown in Fig. 5c and d, respectively. For each layer, the normalized distance to cloud top is calculated as the vertical distance from the point to the cloud top, divided by the total cloud depth, representing the relative vertical location within the layer. High EDR is more likely to appear in the upper part of the cloud under both conditions, which is consistent with the expected effect of cloud-top cooling. When the SCL appears near the base of the cloud, although the median EDR for this situation (Fig. 5c) is still significantly larger than the median EDR of snow without SCL (Fig. 5d) at the cloud base, these SCLs at cloud base are not more likely to be associated with high EDR. The distribution of EDR with SCL near the cloud base suggests that the SCL located below ice or beneath higher SCLs tends to be decoupled from strong turbulence. This suggests that radiative cooling near the cloud top is a stronger factor in generating and maintaining turbulence than latent heat release from SCL formation.

However, for every normalized distance-to-cloud-top height bin, the distribution of EDR under SCL conditions (Fig. 5c) shifts toward higher values compared to that under ISO conditions (Fig. 5d). Although high EDR is more likely to occur in the upper part of the cloud under both conditions, there is a noticeable difference in the vertical extent of the high-frequency, high-EDR region. For ISO conditions, low EDR accounts for more than half of the data from a normalized distance of -0.3 down to the cloud base. For SCL conditions, high EDR is more prevalent throughout the entire upper region of the cloud. This suggests that SCL is correlated with turbulence throughout the cloud.

The median MDV gradient is large (> 0.7 m s<sup>-1</sup> km<sup>-1</sup>) in the SCL conditions as shown in Fig. 2d, and high EDR also distribute in SCL conditions as shown in Fig. 5a. Their distributions almost overlap by comparing the Fig. 2d and Fig 5a. Either of SCL or turbulence are suggested to enhance the ice growth in previous research. To explore the role of turbulence and SCL on ice particle growth respectively, we classify the environments of saturated with respect to ice into six groups: high EDR with SCL, SCL only, high EDR only, ISO with low EDR, high EDR in ice-subsaturated condition, and low EDR in ice-subsaturated condition. The order of the median of MDV gradient and dBZ gradient under ice-supersaturated conditions (Si > 1.0) from large to small are: SCL with high EDR, high EDR only, SCL only, ISO with low EDR (Table 1). That means that from the vertical gradient point of view, turbulence alone can enhance particle growth, which is consistent with previous studies (Chu et al., 2018; Chellini and Kneifel, 2024). The appearance of SCL alone can enhance ice growth, presumably via riming. When turbulence and SCL are coupled together, the MDV gradient and dBZ gradient are much larger than either alone (Table 1), which suggests that the potential positive feedback loop between SCL and turbulence might enhance the growth of ice particles.

In ice-subsaturated conditions (Si < 1.0), a more variable EDR is observed at temperatures above -22°C, varying from  $10^{-9}$  to  $10^{-2}$  m<sup>2</sup> s<sup>-3</sup>. High EDR only appears in warm regions, which likely correspond to lower altitudes, suggesting that turbulence in low Si regions is likely induced by boundary layer processes. As shown in Table 1, the dBZ gradient under high EDR is smaller than that under low EDR in this region, possibly due to turbulence-induced entrainment of dry air and enhanced sublimation in low-relative-humidity environments. In contrast, the MDV gradient is larger in high EDR region. This indicates that while turbulence or entrainment in ice-subsaturated regions (as seen in Fig. 5a) contributes to particle size reduction (i.e., negative dBZ gradient), it may simultaneously enhance particle density (i.e., positive MDV gradient) potentially through morphological changes during sublimation, as reflected in the statistics.

Low EDR is mainly observed near the Si = 1.0 line. At Si = 1.0, ice deposition and sublimation are balanced, stabilizing the local environment and minimizing turbulence. However, when turbulence does occur, it can disrupt this equilibrium by vertically displacing ice particles into regions of differing saturation, enhancing local imbalances.

The RH threshold for identifying SCL and the EDR threshold for identifying high EDR regions were adjusted within a reasonable range (RH: from 95% to 99%, EDR: from  $10^{-6}\,\mathrm{m}^2\,\mathrm{s}^{-3}$  to  $10^{-4}\,\mathrm{m}^2\,\mathrm{s}^{-3}$ ) to verify the robustness of the results above. Although the specific values vary, the main results previously described remain unchanged despite these variations, confirming their reliability.

### 4 Conclusions

This study provides a comprehensive investigation into ice microphysical processes and their interaction with turbulence in Arctic stratiform clouds using 6.5 years of ground-based radar and radiosonde observations at the DOE ARM North Slope of Alaska site. By combining Ka-band radar data with thermodynamic profiles, we leverage radar reflectivity and MDV, along with their vertical gradients to trace the state and the evolution of ice particles under varying temperature and moisture conditions.

The results reveal clear links between water vapor saturation ratio with respect to ice and ice particle evolution: ice supersaturated conditions (Si > 1.0) exhibit positive vertical gradients of radar variables (i.e., reflectivity and MDV), indicating ice particle growth, while ice-subsaturated conditions (Si < 1.0) exhibit negative gradients of radar variables, indicating sublimation. The vertical gradients of reflectivity and mean Doppler velocity directly reflect ice growth processes, making them more closely tied to environmental conditions. Our observational insights provide a potential perspective for evaluating ice-growth processes, which remain a major source of uncertainty in cloud-resolving models. The shape of ice hydrometeors varies greatly due to a range of growth processes influenced by changing environmental conditions during their lifetimes. This variability makes it difficult to accurately simulate their scattering properties and fall speeds (Tyynelä et al., 2023). Instead of relying solely on empirical relationships—such as mass—diameter (M–D) or velocity—diameter (V–D) curves for different ice types (e.g. Locatelli and Hobbs, 1974)—to describe transitions in mass and number concentrations across predefined categories (Grabowski et al., 2019), an alternative approach could model size and density growth rates as functions of the microphysical processes driven by the environment.

Temperature dependence of ice growth is observed in ice supersaturated regions, which is consistent with previous laboratory studies. In the dendritic growth zone (around -15°C), positive dBZ gradients are accompanied by negative Doppler velocity gradients, particularly pronounced for small ice particles (as indicated by low dBZ values). This pattern suggests the transitions from fast-falling to slow-falling ice particles near this temperature, or the emergence of newly-formed, slowly-falling ice particles through secondary ice production. A similar pattern is observed around -22°C, although it is not as significant as that around -15°C.

Turbulence, quantified by EDR, is strongest and most concentrated in SCL regions compared to ISO conditions and ice-subsaturated conditions. Although EDR is relatively larger near cloud top than that below, the correlation between SCL and turbulence can be observed throughout the cloud. The coexistence of SCL and high EDR substantially enhances ice particle growth, as indicated by pronounced vertical gradients. The median vertical gradient of reflectivity in these regions is 12.99 dBZ km<sup>-1</sup>, and the median gradient of MDV is 0.96 m s<sup>-1</sup> km<sup>-1</sup>, both significantly higher than in SCL only regions (8.17 dBZ km<sup>-1</sup> and 0.25 m s<sup>-1</sup> km<sup>-1</sup>, respectively) and turbulence only regions (9.9 dBZ km<sup>-1</sup>, 0.48 m s<sup>-1</sup> km<sup>-1</sup>). While either SCL or high EDR alone can promote ice growth, their individual contributions are 24–37% lower in reflectivity gradient and 50–74% lower in MDV gradient, compared to regions where both coexist. High EDR in ice-subsaturated regions at temperatures above -22°C is possibly driven by boundary layer processes, which tends to reduce particle size but enhance particle density as indicated by vertical gradient of reflectivity and MDV.

Although our observations provide robust statistics, they are limited to primarily stratiform cloud regimes due to the infrequent occurrence of deep convection over the North Slope of Alaska. The role of convective dynamics and associated turbulence in shaping ice microphysics remains an important open question. Future studies targeting convective systems may offer valuable insights. Furthermore, utilizing multi-frequency and dual-polarization radar systems could also improve the characterization of ice particle microstructure and ice microscopical processes, thereby advancing understanding of microphysical complexity (e.g. Kumjian et al., 2022; Tyynelä and Chandrasekar, 2014; Leinonen et al., 2013; Oue et al., 2021; von Terzi et al., 2022).

Our study provides a map of ice particle growth and sublimation—diagnosed from vertical gradients of radar reflectivity and mean Doppler velocity—across a broad range of temperature and moisture conditions based on long-term radar observations. It confirms and extends previous findings on temperature-dependent ice growth from short-term observations and laboratory experiments by providing statistical quantification over a multi-year dataset. In addition, our results enhance understanding of the role of turbulence in ice microphysics and underscore the importance of incorporating both environmental conditions and turbulence to improve the representation of ice microphysical processes in models.

*Author contributions*. Data analysis, post-processing and generation of figures was performed by JY. Conceptualization of the method, interpretation, and writing were shared between JY, MO, PK, EL and FY. Data pre-processing was performed by EL.

320 Competing interests. The contact author has declared that none of the authors has any competing interests.

315

325

Acknowledgements. Jialin Yan and Pavlos Kollias were supported National Aeronautics and Space Administration (NASA) under the Atmospheric Observing System (AOS) project (Contract number: 80NSSC23M0113). Mariko Oue and Pavlos Kollias were supported by the National Science Foundation grant number AGS1841215. Mariko Oue was also supported by DOE grant numbers DE-SC0021160 and DE-SC0025146. Edward Luke, Pavlos Kollias, and Fan Yang were funded by DOE as part of the Atmospheric System Research (ASR) program under contract DE-SC0012704.

Data availability. Data from KAZR radar, radiosonde, and ceilometer (DOIs: https://doi.org/10.5439/1976090, https://doi.org/10.5439/1595321, https://doi.org/10.5439/1181954) are available from the ARM Data Discovery website (https://www.archive.arm.gov/discovery/).

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
