# Peer review of "A radar view of ice microphysics and turbulence in Arctic cloud systems"

_EGUsphere, 2025_

## Author Comment (AC1)

We thank the reviewers for their valuable comments and suggestions. We have addressed all their comments without an exception and made the necessary modifications to the manuscript accordingly. In the following, we provide point-by-point responses to each comment. The comments of the reviewers are in ***bold italics*** and the revised texts are in blue.

**Authors' response to reviewer #1 Peter May**

***1. This paper analyses a large volume of K-band radar data from the North Slope of Alaska ARM site and uses novel statistical methods to infer some key ice microphysical processes from the data. This is well within the scope of ACP and these broad statistical studies are to be encouraged. The analysis appears quite robust and I think this manuscript should be published with relatively minor edits. I also suggest that some discussion is made on the potential use of longer wavelength data given the quite large number of Microwave Rain Radars operating at high latitudes. For scatter from ice particles there should not be too much sensitivity.***

We sincerely thank Dr. Peter May for his positive evaluation and thoughtful comments. We also appreciate the suggestion to include a discussion on the potential use of longer-wavelength radar data.

Our measurements rely on cloud radar, which provides sufficient sensitivity to detect liquid cloud droplets and small ice crystals (i.e., the initial stages of snow particle formation). In contrast, longer-wavelength radars (e.g., X-, C-, and S-band) offer improved penetration through precipitation and are less affected by attenuation, particularly in liquid precipitation. However, they generally lack the sensitivity required to detect small cloud droplets or ice crystals. As a result, such systems may not clearly capture the deceleration in mean Doppler velocity (MDV) under conditions where large precipitation particles coexist with smaller ice crystals.

We have now included a discussion of this limitation in the Introduction. Additionally, we agree that incorporating longer-wavelength radar observations could provide valuable multi-frequency datasets, which may enhance the retrieval of ice particle shape and support improved understanding of ice microphysical processes. This potential benefit is now mentioned in Section 4.

"KAZR provides much higher sensitivity to small cloud droplets and ice crystals, allowing clear detection of MDV deceleration when large precipitation particles coexist with smaller

hydrometeors, compared to longer-wavelength radars (X, C, S bands) (Kollias et al., 2020).
"

"Furthermore, utilizing multi-frequency and dual-polarization radar systems could also improve the characterization of ice particle microstructure and ice microscopical processes, thereby advancing understanding of microphysical complexity (e.g. Kumjian et al., 2022; Tyynelä and Chandrasekar, 2014; Leinonen et al., 2013; Oue et al., 2021; von Terzi et al., 2022). "

*2. There are a couple of gaps in the discussion. The introduction of Z being proportional to the sixth power of D is true for Rayleigh scatter, but scattering from ice crystals is much more complex with dependencies on shape, density and how much air is trapped than such a simple relation. There is an excellent discussion in Chapter 3 by Tynelä et al in the recent book, Volume 2 of Advances in Weather Radar edited by V.N Bringi, K.V. Mishra and M. Thurai. Likewise, the variations in (reflectivity weighted) fall speed for different crystals and the impact of this on the discussion and interpretation needs some further discussion. This discussion goes back a long way, at least to Locatelli and Hobbs (JGR, 1974). This is mentioned in the latter part of the manuscript, but again needs some more detail and nuancing.*

We thank for highlighting this point. We have given more discussion and revised the relevant sections in section 2, sections 3, and section 4, and included Tyynelä et al. (Chapter 3, Volume 2 of *Advances in Weather Radar*) and Locatelli and Hobbs (JGR, 1974) in references.

"Under the assumption of Rayleigh scattering, reflectivity is proportional to number concentration and the sixth power of diameter ($D^6$), thereby reflecting particle size and/or number. In addition, the highly complex shapes of ice particles and the potential existence of Mie scattering make the simulation of scattering for ice particles difficult (Tyynelä et al., 2023)."

"Except for the size or density growth rate, the particle shape transition may also be a significant source of change of scattering properties, like reflectivity, as revealed by scattering simulations (Tyynelä et al., 2023, and references therein)."

"The shape of ice hydrometeors varies greatly due to a range of growth processes influenced by changing environmental conditions during their lifetimes. This variability makes it difficult to accurately simulate their scattering properties and fall speeds (Tyynelä et al., 2023). Instead of relying solely on empirical relationships—such as mass–diameter (M–D)

or velocity–diameter (V–D) curves for different ice types (e.g. Locatelli and Hobbs, 1974)—to describe transitions in mass and number concentrations across predefined categories (Grabowski et al., 2019), an alternative approach could model size and density growth rates as functions of the microphysical processes driven by the environment."

***3. The discussion on EDR retrievals also needs further explanation and what equations are being used? It is certainly different from spectral width based estimates. What is the confidence in these retrievals?***

Thank you for pointing this out. We agree that the current explanation of EDR retrievals needs further detail. In the revised manuscript (Section 2.3), we mention the advantage of the EDR retrievals used in this study, and include the equation and uncertainty.

"The eddy dissipation rate (EDR), used to characterize turbulence intensity, is calculated from mean Doppler velocity observations from KAZR and horizontal wind data from radiosondes following the velocity time-series variance method described in Borque et al. (2016), which provides robust estimates in cases where the spread of the particle size distribution significantly affects spectrum width. As implemented here, a 30-minute window centered on the radiosonde profile is used to extract mean Doppler velocity time series at each altitude. If fewer than 300 mean Doppler velocity measurements are available to consist a time series within this window, EDR is not computed for that altitude, as shown in Fig. 1(d). As detailed in Borque et al. (2016), these time series are Fourier-transformed to obtain the velocity spectrum. EDR is then estimated by integrating the spectrum over multiple subranges within the inertial subrange using scaling relations derived from Kolmogorov's theory based on the following equation:

$$EDR = \frac{2\pi}{V_h} \left( \frac{2}{3\alpha} \int_{f_{\text{low}}}^{f_{\text{high}}} S(f)\, df \right) \left( f_{\text{low}}^{-2/3} - f_{\text{high}}^{-2/3} \right)^{-3/2} \tag{1}$$

where $f_{\text{low}}$ and $f_{\text{high}}$ represent the lower and upper bounds of the inertial subrange, $V_h$ is the horizontal wind speed, $\alpha$ is the Kolmogorov constant, and S(f) is the turbulent energy spectrum as a function of frequency. To identify valid inertial subranges, several predefined frequency intervals are tested, and a power-law fit is performed for each. Intervals with spectral slopes within –5/3 ± 1/3 are retained as the intervals in inertial subrange, and the final EDR is computed as the average over these accepted estimates. The uncertainty measured by the standard deviation of the accepted EDR estimates is around 30%, which in $log_{10}$(EDR) corresponds to only around 0.1 dex."

***4. How robust is the "detection" of SCL? For samples where you argue that***

*there is SCL near cloudbase, have you validated with lidar data? This would give more confidence to the conclusions.*

We appreciate the reviewer's suggestion. In our study, we did use ceilometer observations to identify cloud base heights. The close agreement between the cloud base detected by the ceilometer and the base of the inferred SCL layers from radiosonde (RH greater than 95%) serves as a validation of our identification. We have clarified this point in the revised manuscript (Section 2.2). In addition, this method was also verified in previous study by Silber et al. (2021) and Luke et al. (2021).

"Previous research indicates that radiosonde-observed high relative humidity is consistent with the presence of supercooled liquid droplets (SCL) at subfreezing temperatures. For example, Silber et al. (2021) used a threshold of RH above 95% to identify the SCL layer across a wide temperature range, while Luke et al. (2021) used a higher threshold of about 98% around -5 °C. In this study, we use the relative humidity threshold of 95% to indicate the potential existence of SCL, as it provides a broadly applicable criterion across the temperature range."

"As shown in Fig. 1, the cloud base heights observed by the ceilometer, which is the lowest altitude of liquid, exhibit good agreement with the base of the first SCL layer above the surface, identified by the radiosonde observations (i.e., RH exceeding 95%). This agreement supports our identification of SCL, with the ceilometer serving as an independent validation on the vertical placement of SCL inferred from radiosonde measurements."

*5. I certainly wouldn't expect that turbulence contributes to the formation of SCL. In contrast, I thought it would increase collision rates and riming.*

Thank you for sharing your thoughts. We see your points that turbulence is associated with increased collision rates and riming. Previous studies have shown that turbulence can contribute to the formation of SCL (Morrison et al., 2012; Korolev and Field, 2008). For example, Korolev and Field (2008) suggested that mixed-phase clouds can also form by generating supercooled liquid layer inside an ice cloud through vertical air motion, in contrast to the conventional idea of generating ice particles via ice nucleation inside a pure supercooled liquid cloud. From our observation in Fig. 5a, within the SCL region, the median EDR is higher than that under other humidity conditions. A closer examination of the distribution of SCL and EDR in a cloud Fig. 5c-d) reveals that, for every normalized distance-to-cloud-top height bin, the EDR under SCL conditions consistently shifts toward higher values compared to that under ISO conditions (ice saturated only, RH < 95% and Si > 1). We believe it can suggest that there is some relations between turbulence and SCL

throughout the cloud. We have removed the unclear sentence to avoid confusion and reprase the related part to make it clearer.

"As shown in Fig. 5a, most of the medians of the EDR in SCL conditions are of values greater than $10^{-5}\,\mathrm{m^2\,s^{-3}}$, and larger than those in ISO and Si $< 1$ conditions. The PDFs of EDR also show that EDR values in SCL conditions are relatively larger, suggesting stronger turbulence therein (Fig. 5b). The strong correlation between high EDR and the SCL condition may be related to multiple processes. On the one hand, liquid droplet formation and growth in the mixed-phase clouds can enhance turbulence via the latent heat release. In addition, stronger radiative cooling above the liquid cloud top can also enhance turbulence therein (e.g. Lonardi et al., 2024). On the other hand, turbulence may play an active role in maintaining humidity conditions favorable for the SCL formation (Morrison et al., 2012; Korolev and Field, 2008), potentially creating a positive feedback. "

"We further analyze the normalized frequency of the EDR as a function of normalized distance from cloud top under SCL and ISO conditions, as shown in Fig. 5c and d, respectively. For each layer, the normalized distance to cloud top is calculated as the vertical distance from the point to the cloud top, divided by the total cloud depth, representing the relative vertical location within the layer. High EDR is more likely to appear in the upper part of the cloud under both conditions, which is consistent with the expected effect of cloud-top cooling. When the SCL appears near the base of the cloud, although the median EDR for this situation (Fig. 5c) is still significantly larger than the median EDR of snow without SCL (Fig. 5d) at the cloud base, these SCLs at cloud base are not more likely to be associated with high EDR. The distribution of EDR with SCL near the cloud base suggests that the SCL located below ice or beneath higher SCLs tends to be decoupled from strong turbulence. This suggests that radiative cooling near the cloud top is a stronger factor in generating and maintaining turbulence than latent heat release from SCL formation."

"However, for every normalized distance-to-cloud-top height bin, the distribution of EDR under SCL conditions (Fig. 5c) shifts toward higher values compared to that under ISO conditions (Fig. 5d). Although high EDR is more likely to occur in the upper part of the cloud under both conditions, there is a noticeable difference in the vertical extent of the high-frequency, high-EDR region. For ISO conditions, low EDR accounts for more than half of the data from a normalized distance of $-0.3$ down to the cloud base. For SCL conditions, high EDR is more prevalent throughout the entire upper region of the cloud. This suggests that SCL is correlated with turbulence throughout the cloud."

*6. Do you make a density correction for the fall speeds? This will be needed for more quantitative discussion.*

Yes, we did. We confirm that an air density correction has already been applied to all MDV in the manuscript except for the MDV in the case (Fig. 1), as described in previous Line 92. We have revised the sentence a little bit to make it clearer.

"To account for the effect of air density on fall speed, a density correction was applied to the MDV prior to statistical analysis."

**7. The colorscale of Fig 1, panel b should be changed so that detail between 0 and 1 m/s is more clearly visible.**

We accept this advice. The revised figure is at the end of the document.

**Authors' response to reviewer #2**

*General comments to the manuscript*

*In the manuscript titled "A radar view of ice microphysics and turbulence in Arctic stratiform cloud systems" by J. Yan et al., the authors present a study on ice microphysics and turbulence in Arctic stratiform clouds, based on 6.5 years of Ka-band radar and radiosonde observations at the DOE ARM North Slope of Alaska site. The research focuses on understanding ice particle growth and sublimation processes (via tracking of vertical gradients of radar reflectivity and mean Doppler velocity), their temperature and moisture dependencies, and the role of turbulence in these processes.*

*Recommendation:*

*I would suggest the manuscript to be published after minor revisions considering the remarks below.*

We sincerely thank the anonymous reviewer # 2 for the positive overall evaluation and the constructive and thoughtful comments. Below we provide point-by-point responses to each of the suggestions. We also revised the manuscript taking into account the comments from Dr. Peter May. Please also see our responses to his comments.

*General/Major comments:*

*1. The literature study in the introduction should be extended to acknowledge further studies on secondary ice production (SIP, the process should be explained*

*first, e.g. https://doi.org/10.5194/acp-20-11767-2020) as well as Hallett-Mossop ice splintering (also explain the process and add references) and also acknowledge studies that studied the influence of environmental conditions affecting SIP (e.g. https://doi.org/10.5194/acp-20-1391-2020, https://doi.org/10.5194/acp-21-14671-2021 among others).*

We thank the reviewer for this helpful suggestion. We have revised the introduction to include the a short explanation of secondary ice production (SIP) and Hallett-Mossop (HM) process, and cite relevant studies, including the ones you recommended.

" Secondary ice production (SIP), the formation of atmospheric ice requiring preexisting ice particles, is also a key microphysical process in clouds (e.g. Kanji et al., 2017; Korolev and Leisner, 2020). More than six mechanisms have been identified through laboratory and field studies, but the Hallett–Mossop riming-splintering process remains the only one widely represented in models, typically occurring between -3 and -8 °C (Korolev and Leisner, 2020). Recent remote sensing observational evidence also suggests that secondary ice production frequently occurs at temperatures above -10°C (Kumjian et al., 2020; Luke et al., 2021; Oue et al., 2018; Korolev et al., 2020)."

*2. Please make sure you use precise and consistent wording throughout the manuscript. E.g., on line 42 you state that you want to characterize the two ice processes depositional growth and sublimation while later on you also refer to aggregation and riming as ice microphysical processes. Please also try to improve readability in the result section as indicated by specific comments below.*

We have carefully gone through all your specific comments and revised the manuscript accordingly as shown below. We truly appreciate the effort you put into identifying numerous detailed issues related to wording consistency and clarity throughout the manuscript. For the specific question you mention, we have replaced the "i.e." to "e.g." on line 42.

"The objective of our study is to characterize ice microphysical processes (e.g., depositional growth and sublimation) and in-cloud turbulence, based on long-term KAZR observations under a wide range of environmental conditions derived from radiosondes and examine their relationships. "

*3. The title states that only stratiform cloud systems are considered. I could not find a section that explains how convective clouds are filtered from the data set. – Please clarify/change title.*

Thank you for this comment. In the original title, we referred to stratiform cloud systems to emphasize the typical cloud characteristics in the Arctic. However, we agree that simply referring to "Arctic clouds" is more concise and better aligned with the scope of our study. We have therefore revised the title accordingly.

*Minor Comments:*

*Line 22: replace "vapors" with "water vapor"*

Done.

*Line 37-38: Unclear sentence, please rephrase. What do you consider as limitations of the dataset used in Chellini and Kneifel, 2024?*

We have rephrased the sentence to make it clear.

"Recent statistical analysis from 15-months of radar observations has revealed that the increased turbulence, quantified by the eddy dissipation rate (EDR), enhances aggregation and riming in Arctic low-level mixed-phase clouds (Chellini and Kneifel, 2024). However, this analysis does not account for the influence of humidity and temperature at each observation point, which may also play an important role."

*Line 40: remove "s" from multi-years*

Done.

*Line 41 (and elsewhere): replace radiosonde with radiosondes or "radiosonde observations" and add ceilometer (used in Fig 1a)*

Done.

*Line 48: Add an explanation why you limit your dataset to this specific time frame Jan 2013 – May 2019 instead of extending it to more recent data.*

"The observations in this study are collected from the DOE ARM NSA atmospheric observatory from January 2013 to May 2019, which is the same dataset as used by Luke et al. (2021). This dataset provides a continuous and sufficiently long record for deriving robust statistics."

*Line 50: reference missing*

Done.

*Line 62: after "grow" add "by water vapor deposition"*

Done.

*Line 66: Clarify what is meant by "overall".*

We have added more text to clarify this point.

"Previous research indicates that radiosonde-observed high relative humidity is consistent with the presence of supercooled liquid droplets (SCL) at subfreezing temperatures. For example, Silber et al. (2021) used a threshold of RH above 95% to identify the SCL layer across a wide temperature range, while Luke et al. (2021) used a higher threshold of about 98% around -5 °C. In this study, we use the relative humidity threshold of 95% to indicate the potential existence of SCL, as it provides a broadly applicable criterion across the temperature range."

*Line 66-71: This paragraph should acknowledge that besides high relative humidities, cloud condensation nuclei are required for the formation of liquid droplets. Also, consider shading the three defined moisture regimes in Fig 1 c).*

We accept the suggestion to add shading to show the three defined moisture regimes in Fig. 1c and also revised the sentence as below. The revised figure is at the end of the document.

"We define three moisture regimes: SCL conditions (RH > 95%, Si > 1.0), which favors the existence of SCL, considering sufficient cloud condensation nuclei are available; ISO (ice supersaturated only) conditions (RH < 95%, Si > 1.0), where deposition is possible but the occurrence of liquid water is highly unlikely; and ice-subsaturated conditions (Si < 1.0), where neither liquid water nor depositional growth is expected."

*Line 74: remove "And" at beginning of sentence, mixed-phase (instead of mix-phase)*

Done.

*Line 89: remove "or"*

Done.

*Line 90-92: Add that this assumption of negligible vertical air motion can be made for stratiform Arctic clouds but not e.g. deep convective systems and give*

*references for other studies where this assumption has been used. Please also state at which altitude you start calculating the gradients – from radar echo top downwards or a certain height within the clouds?*

Thanks for the suggestion. We have modified the manuscript accordingly.

"Assuming that vertical air motion is negligible relative to particle fall speed in large statistical samples for Arctic clouds, we interpret the median MDV values as representative of hydrometeor fall speeds, similar to the assumption used in Kalesse et al. (2013)."

"To avoid uncertainties at the cloud boundaries, the gradients are evaluated from the third range gate below the cloud top down to the third range gate above the cloud base."

*Line 140: "for" temperatures instead of "with" temperatures*

Done.

*Line 142: remove sentence as it has same content as the one on line 140*

Done.

*Line 146: typo "riming"*

Corrected.

*Line 146-148: sentence unclear, please rephrase*

The sentence is rephrased.

"At around -15 $^o$C, the size of ice particles may be related to their residence time in favorable conditions. This is because their slow fall speed, as suggested by the low MDV, allows them to persist longer in favorable environments."

*Line 149-154: Please also describe the MDV decrease for the SCL condition in the dendritic growth zone around -15 C – it is even more pronounced than for ISO conditions.*

We have revised sentence.

"The MDV reaches its local maximum value at about -6 °C, while it attains its local minimum value near -15 °C for both SCL and ISO conditions. The decrease in MDV for the SCL condition is more pronounced than that for the ISO condition."

*Line 159: Since you are explaining ISO conditions, do you mean Fig 4c) instead of a)?*

The reviewer is correct. We have revised it based on the suggestion.

*Line 155 – 169: Please improve the readability of this paragraph. Also, can you explain the other features shown in Fig 4 such as in Fig 4a) negative dBZ gradient for T > -10 C and small reflectivities; Fig 4b) strong positive MDV gradients for T -10 to -20 C and high reflectivities; Fig 4c+d: for T > -10C negative dBZ gradient and positive MDV gradient; Fig 4d: strong positive MDV gradients for T between -25 to -16 C and high reflectivities*

We have added descriptions of the features mentioned by the reviewer and provided explanations where possible, as detailed below. For those aspects that we are currently unable to explain, we hope that model-based studies can provide further insights in the future.

For Fig 4a) negative dBZ gradient for T > -10 C and small reflectivities; Fig 4c+d: for T > -10C negative dBZ gradient and positive MDV gradient:

"Both SCL and ISO conditions exhibit a negative reflectivity gradient accompanied by a positive MDV gradient at -5°C when reflectivity is below -20 dBZ. It suggests that there are fewer slow-falling particles (either cloud droplets or small ice particles) in the current radar range gate than in the gate above, which can occur near the base of tenuous supercooled liquid layers.

Fig 4b) strong positive MDV gradients for T -10 to -20 C and high reflectivities:

Under SCL conditions, the MDV gradient is strongly positive for regions of high reflectivity at temperatures between –10 and –20°C. This feature may be explained by the presence of relatively large and fast-falling snow particles indicated by large reflectivity, which enhances riming and aggregation. In addition, its coincidence with the temperature range of –10 to –20 °C further supports the aggregation, as dendrite in this regime are known to aggregate more efficiently (Lamb and Verlinde, 2011).

Fig 4d: strong positive MDV gradients for T between -25 to -16 C and high reflectivities:

Under ISO conditions, strongly positive MDV gradients are observed at high reflectivity within the temperature range of –16 to –25°C. According to Fukuta and Takahashi (1999), the fall speed of ice particles arrives a local minimum at around -22°C and a local maximum at around -20°C when the growth time is sufficiently long, which may explain the positive MDV gradient at high reflectivity around that temperature range. "

*Line 166-167: unclear sentence, please rephrase*

The sentence is rephrased.

"For both SCL and ISO conditions, small or negative MDV gradients coexist with large dBZ gradients at relatively low reflectivity around -15 °C, suggesting that there are additional slow-falling ice particles at one height than the height above. This is because at low reflectivity (inferring small preexisting particles), an increase in the number of slow-falling particles contributes substantially to reflectivity and also reduces the MDV, leading to increased dBZ and decreased MDV gradients. In contrast, when preexisting particles are already large (high reflectivity), size effects from preexisting particles dominate over number concentration, so additional small particles produce only a limited increase in reflectivity and exert little influence on MDV (reflectivity-weighted mean), resulting in smaller gradient change than for the low-reflectivity situations."

*Line 202+203: typo ice-subsaturated*

Done.

*Line 232-239: unclear sentences, please rephrase*

We have adjusted the sentence order in the section and rephrased the unclear sentences to improve clarity, as shown below.

"As shown in Fig. 5a, most of the medians of the EDR in SCL conditions are of values greater than $10^{-5}\,\mathrm{m^2\,s^{-3}}$, and larger than those in ISO and Si $< 1$ conditions. The PDFs of EDR also show that EDR values in SCL conditions are relatively larger, suggesting stronger turbulence therein (Fig. 5b). The strong correlation between high EDR and the SCL condition may be related to multiple processes. On the one hand, liquid droplet formation and growth in the mixed-phase clouds can enhance turbulence via the latent heat release. In addition, stronger radiative cooling above the liquid cloud top can also enhance turbulence therein (e.g. Lonardi et al., 2024). On the other hand, turbulence may play an active role in maintaining humidity conditions favorable for the SCL formation (Morrison et al., 2012; Korolev and Field, 2008), potentially creating a positive feedback. "

"We further analyze the normalized frequency of the EDR as a function of normalized distance from cloud top under SCL and ISO conditions, as shown in Fig. 5c and d, respectively. For each layer, the normalized distance to cloud top is calculated as the vertical distance from the point to the cloud top, divided by the total cloud depth, representing the relative vertical location within the layer. High EDR is more likely to appear in the upper part of

the cloud under both conditions, which is consistent with the expected effect of cloud-top cooling. When the SCL appears near the base of the cloud, although the median EDR for this situation (Fig. 5c) is still significantly larger than the median EDR of snow without SCL (Fig. 5d) at the cloud base, these SCLs at cloud base are not more likely to be associated with high EDR. The distribution of EDR with SCL near the cloud base suggests that the SCL located below ice or beneath higher SCLs tends to be decoupled from strong turbulence. This suggests that radiative cooling near the cloud top is a stronger factor in generating and maintaining turbulence than latent heat release from SCL formation."

"However, for every normalized distance-to-cloud-top height bin, the distribution of EDR under SCL conditions (Fig. 5c) shifts toward higher values compared to that under ISO conditions (Fig. 5d). Although high EDR is more likely to occur in the upper part of the cloud under both conditions, there is a noticeable difference in the vertical extent of the high-frequency, high-EDR region. For ISO conditions, low EDR accounts for more than half of the data from a normalized distance of $-0.3$ down to the cloud base. For SCL conditions, high EDR is more prevalent throughout the entire upper region of the cloud. This suggests that SCL is correlated with turbulence throughout the cloud."

***Comments on Figures:***

***Fig 1: add which time is shown on x-axis (UTC?). in Panel c) and d) please add the SCL bases and tops as done in panel a) and b).***

We now add "UTC" in the Fig. 1 x axis and add SCL bases and tops in Panel c) and d). Revised Fig. 1 is shown at the end of the document.

***Fig 3: In the caption add the what the dashed red line "DI median" refers to (or replace by "ISO" ?). Fig 5b: Replace "DI" with "ISO" in the legend***

The reviewer is correct. We now have replaced "DI" to "ISO" for both of Fig.3 and Fig. 5b as shown at the end of the document.

**References**

Paloma Borque, Edward Luke, and Pavlos Kollias. On the unified estimation of turbulence eddy dissipation rate using doppler cloud radars and lidars. *Journal of Geophysical Research: Atmospheres*, 121(10): 5972–5989, 2016. doi: https://doi.org/10.1002/2015JD024543. URL https://agupubs.onlinelibrary.wiley.com/doi/abs/10.1002/2015JD024543.

Norihiko Fukuta and Tsuneya Takahashi. The growth of atmospheric ice crystals: A summary of findings in vertical supercooled cloud tunnel studies. *Journal of the Atmospheric Sciences*, 56(12):1963–1979, 1999. ISSN 0022-4928. doi: https://doi.org/10.1175/1520-0469(1999)056<1963:TGOAIC>2.0.CO;2. URL `https://journals.ametsoc.org/view/journals/atsc/56/12/1520-0469`$_1$`999`$_0$`56`$_1$`963`$_t$`goaic`$_2$`.0.co`$_2$`.xn`

Wojciech W Grabowski, Hugh Morrison, Shin-Ichiro Shima, Gustavo C Abade, Piotr Dziekan, and Hanna Pawlowska. Modeling of cloud microphysics: Can we do better? *Bulletin of the American Meteorological Society*, 100(4):655–672, 2019.

Heike Kalesse, Pavlos Kollias, and Wanda Szyrmer. On using the relationship between doppler velocity and radar reflectivity to identify microphysical processes in midlatitudinal ice clouds. *Journal of Geophysical Research: Atmospheres*, 118(21):12,168–12,179, 2013. ISSN 2169-897X. doi: https://doi.org/10.1002/2013JD020386. URL `https://agupubs.onlinelibrary.wiley.com/doi/abs/10.1002/2013JD020386`.

Zamin A. Kanji, Luis A. Ladino, Heike Wex, Yvonne Boose, Monika Burkert-Kohn, Daniel J. Cziczo, and Martina Krämer. Overview of Ice Nucleating Particles. 58:1.1–1.33, 2017. ISSN 0065-9401. doi: 10.1175/AMSMONOGRAPHS-D-16-0006.1. URL `http://journals.ametsoc.org/doi/10.1175/AMSMONOGRAPHS-D-16-0006.1`.

P. Kollias, N. Bharadwaj, E. E. Clothiaux, K. Lamer, M. Oue, J. Hardin, B. Isom, I. Lindenmaier, A. Matthews, E. P. Luke, S. E. Giangrande, K. Johnson, S. Collis, J. Comstock, and J. H. Mather. The arm radar network: At the leading edge of cloud and precipitation observations. *Bulletin of the American Meteorological Society*, 101(5):E588–E607, 2020. ISSN 0003-0007. doi: https://doi.org/10.1175/BAMS-D-18-0288.1. URL `https://journals.ametsoc.org/view/journals/bams/101/5/bams-d-18-0288.1.xml`.

A. Korolev and T. Leisner. Review of experimental studies of secondary ice production. *Atmos. Chem. Phys.*, 20(20):11767–11797, 2020. ISSN 1680-7324. doi: 10.5194/acp-20-11767-2020. URL `https://acp.copernicus.org/articles/20/11767/2020/`. ACP.

A. Korolev, I. Heckman, M. Wolde, A. S. Ackerman, A. M. Fridlind, L. A. Ladino, R. P. Lawson, J. Milbrandt, and E. Williams. A new look at the environmental conditions favorable to secondary ice production. *Atmos. Chem. Phys.*, 20 (3):1391–1429, 2020. ISSN 1680-7324. doi: 10.5194/acp-20-1391-2020. URL `https://acp.copernicus.org/articles/20/1391/2020/`. ACP.

Alexei Korolev and Paul R Field. The effect of dynamics on mixed-phase clouds: Theoretical considerations. *Journal of the Atmospheric Sciences*, 65(1):66–86, 2008.

Matthew R. Kumjian, Daniel M. Tobin, Mariko Oue, and Pavlos Kollias. Microphysical insights into ice pellet formation revealed by fully polarimetric ka-band doppler radar. *Journal of Applied Meteorology and Climatology*, 59(10):1557–1580, 2020. doi: 10.1175/JAMC-D-20-0054.1. URL https://journals.ametsoc.org/view/journals/apme/59/10/jamcD200054.xml.

Matthew R. Kumjian, Olivier P. Prat, Karly J. Reimel, Marcus Van Lier-Walqui, and Hughbert C. Morrison. Dual-Polarization Radar Fingerprints of Precipitation Physics: A Review. *Remote Sensing*, 14(15):3706, August 2022. ISSN 2072-4292. doi: 10.3390/rs14153706.

Dennis Lamb and Johannes Verlinde. *Physics and Chemistry of Clouds*. Cambridge University Press, 2011. ISBN 978-0-511-97637-7 978-1-139-19063-3 978-1-62870-283-5.

J. Leinonen, D. Moisseev, and T. Nousiainen. Linking snowflake microstructure to multi-frequency radar observations. 118(8):3259–3270, 2013. ISSN 2169-897X, 2169-8996. doi: 10.1002/jgrd.50163. URL https://agupubs.onlinelibrary.wiley.com/doi/10.1002/jgrd.50163.

John D. Locatelli and Peter V. Hobbs. Fall speeds and masses of solid precipitation particles. *Journal of Geophysical Research*, 79(15):2185–2197, 1974. ISSN 0148-0227. doi: 10.1029/jc079i015p02185. URL https://dx.doi.org/10.1029/jc079i015p02185. FIg 10-43 and 10-44.

M. Lonardi, E. F. Akansu, A. Ehrlich, M. Mazzola, C. Pilz, M. D. Shupe, H. Siebert, and M. Wendisch. Tethered balloon-borne observations of thermal-infrared irradiance and cooling rate profiles in the Arctic atmospheric boundary layer. *Atmospheric Chemistry and Physics*, 24(3):1961–1978, 2024. doi: 10.5194/acp-24-1961-2024.

Edward P. Luke, Fan Yang, Pavlos Kollias, Andrew M. Vogelmann, and Maximilian Maahn. New insights into ice multiplication using remote-sensing observations of slightly supercooled mixed-phase clouds in the arctic. *Proceedings of the National Academy of Sciences*, 118(13):e2021387118, 2021. ISSN 0027-8424. doi: 10.1073/pnas.2021387118. URL https://dx.doi.org/10.1073/pnas.2021387118.

Hugh Morrison, Gijs De Boer, Graham Feingold, Jerry Harrington, Matthew D. Shupe, and Kara Sulia. Resilience of persistent arctic mixed-phase clouds. *Nature Geoscience*, 5(1):11–17, 2012. ISSN 1752-0894. doi: 10.1038/ngeo1332. URL https://dx.doi.org/10.1038/ngeo1332. SCL and turbulence Relation Fig 2 a.

Mariko Oue, Pavlos Kollias, Alexander Ryzhkov, and Edward P. Luke. Toward exploring the synergy between cloud radar polarimetry and doppler spectral analysis in deep cold precipitating systems in the arctic. *Journal of Geophysical Research: Atmospheres*, 123(5):2797–2815, 2018. ISSN 2169-897X. doi: 10.1002/2017jd027717. URL https://dx.doi.org/10.1002/2017jd027717.

Mariko Oue, Pavlos Kollias, Sergey Y. Matrosov, Alessandro Battaglia, and Alexander V. Ryzhkov. Analysis of the microphysical properties of snowfall using scanning polarimetric and vertically pointing multi-frequency Doppler radars. 14 (7):4893–4913, 2021. ISSN 1867-8548. doi: 10.5194/amt-14-4893-2021. URL https://amt.copernicus.org/articles/14/4893/2021/.

Israel Silber, Paul S. McGlynn, Jerry Y. Harrington, and Johannes Verlinde. Habit-dependent vapor growth modulates arctic supercooled water occurrence. *Geophysical Research Letters*, 48(10), 2021. ISSN 0094-8276. doi: 10.1029/2021gl092767. URL https://dx.doi.org/10.1029/2021gl092767.

J. Tyynelä and V. Chandrasekar. Characterizing falling snow using multifrequency dual-polarization measurements: TYYNELÄ AND CHANDRASEKAR. *Journal of Geophysical Research: Atmospheres*, 119(13):8268–8283, July 2014. ISSN 2169897X. doi: 10.1002/2013JD021369.

Jani Tyynelä, Karina McCusker, Davide Ori, Robin Ekelund, and Ines Fenni. Scattering by snow particles. In V.N. Bringi, Kumar Vijay Mishra, and Merhala Thurai, editors, *Advances in Weather Radar. Volume 2: Precipitation Science, Scattering and Processing Algorithms*, pages 99–142. Institution of Engineering and Technology, 2023. ISBN 978-1-83953-624-3 978-1-83953-625-0. doi: $10.1049/SBRA557G_ch3.URL$

[Figure]

Figure 1: A case of mixed-phase clouds observed on February 18, 2015 at the NSA site. (a) Radar reflectivity and (b) Mean Doppler Velocity (MDV) from KAZR within ±15 minutes and 4 km of the radiosonde launch, plotted as time–height sections in UTC. (c) Radiosonde profiles of relative humidity and water vapor saturation ratio with respect to ice (Si). (d) In-cloud energy dissipation rate (EDR) from radiosonde and radar. In (a–b), dashed purple lines mark the top and bottom boundaries of radiosonde-identified supercooled liquid water (SCL) layers; for clarity, two closely spaced SCL layers separated by a thin non-SCL interval are observed. Black points denote ceilometer-identified cloud base height (Zhang et al., 1997). In (c), the vertical dashed red line marks RH = 95% (SCL threshold), and the vertical dashed blue line marks Si = 1. Colored shadings indicate moisture regimes: blue for the SCL condition (RH > 95%, Si > 1.0), purple for the ISO condition (RH < 95%, Si > 1.0), and orange for the ice-subsaturated condition (Si < 1.0).

[Figure]

Figure 3: Normalized frequency distribution of reflectivity (dBZ, panels a-b) and mean Doppler velocity (MDV, m s$^{-1}$, panels c-d) as a function of temperature (°C) under different moisture conditions. Panels (a) and (b) correspond to the SCL condition, while panels (c) and (d) correspond to the ISO condition. The dashed black and red lines represent the median profiles for the SCL condition and ISO condition, respectively. The blue dashed line indicates the reference line for MDV = 0 m s$^{-1}$. The color shading indicates the normalized frequency of observations within each temperature-MDV or temperature-reflectivity bin.

[Figure]

Figure 5: (a) Median values of EDR as a function of relative humidity (%) and temperature (°C), the black contour lines indicate different saturation levels with respect to ice (Si = 1.25, 1.0, and 0.75); (b) PDF of EDR under SCL conditions (blue line), ISO conditions (orange line), and Si < 1.0 conditions (green line). Vertical distribution of in-cloud EDR under (c) SCL conditions and (d) ISO conditions.